# New Sorbicillinoids from the Mangrove Endophytic Fungus *Trichoderma reesei* SCNU-F0042

**DOI:** 10.3390/md21080442

**Published:** 2023-08-05

**Authors:** Jialin Li, Tao Chen, Jianchen Yu, Hao Jia, Chen Chen, Yuhua Long

**Affiliations:** 1GDMPA Key Laboratory for Process Control and Quality Evaluation of Chiral Pharmaceuticals, School of Chemistry, South China Normal University, Guangzhou 510006, China; jialinli@m.scnu.edu.cn (J.L.); chent296@mail2.sysu.edu.cn (T.C.); haojia@m.scnu.edu.cn (H.J.); chenchen2021@m.scnu.edu.cn (C.C.); 2Key Laboratory of Tropical Disease Control, Ministry of Education, Department of Biochemistry, Zhongshan School of Medicine, Sun Yat-Sen University, Guangzhou 510080, China; yujch3@mail.sysu.edu.cn

**Keywords:** mangrove endophytic fungus, *Trichoderma reesei*, sorbicillinoids, SARS-CoV-2 inhibitory activity, molecular networking

## Abstract

Three new dimeric sorbicillinoids (**1**–**3**) and one new 3,4,6-trisubstituted α-pyrone (**5**), along with seven analogues (**4** and **6**–**11**), were isolated from the mangrove endophytic fungus *Trichoderma reesei* SCNU-F0042 under the guidance of molecular networking approach. Their chemical structures were established by 1D and 2D NMR HR-ESI-MS and ECD analysis. In a bioassay, compound **2** exhibited moderate SARS-CoV-2 inhibitory activity with an EC_50_ value of 29.0 μM.

## 1. Introduction

*Trichoderma* sp. is a widespread filamentous fungus of ascomycetes in various types of soils [1]. Among them, the mangrove-derived fungal genus *Trichoderma* produces a diverse array of secondary metabolites including alkaloids, polyketides, terpenoids, phenols, lactones, and various hybrids of the aforementioned classes [2,3,4,5,6]. Sorbicillinoids, a marker secondary metabolite of *Trichoderma reesei* (*T*. *reesei*), have a characteristic sorbyl side chain and a cyclic hexaketide nucleus in the structures [7,8]. Since sorbicillin was first discovered from *Penicillium notatum* in 1948, more than 100 analogs of sorbicillinoids have been reported and they can be classified into monomeric-, bi-, tri-, and hybrid sorbicillinoids by the number of sorbicillinoid construction units [9,10]. Many of them exhibited a wide range of biological activities, such as cytotoxic [11], antibacterial [12], antifungal [13], anti-inflammatory [14], phytotoxic [15], and *α*-glucosidase inhibitory activity [16].

Molecular networking, a strategy that organizes and analyses MS/MS data based on chemical similarity, can be used for dereplication in natural products discovery [17]. After extraction and concentration, the EtOAc extract was subjected to LC-MS/MS analysis, then the data were uploaded to the Global Natural Product Social Molecular Networking (GNPS; www.gnps.ucsd.edu) platform, followed by MN analysis using the online workflow. A comprehensive examination of the MS^2^ spectra libraries allowed the annotation of the node at *m*/*z* 497.217 (C_28_H_33_O_8_, [M + H]^+^) as the bislongiquinolide (**4**) based on its fragmentation pattern [18]. By the guidance of bislongiquinolide node, more new bioactive sorbicillinoids remain to be discovered in the metabolites from the fungus *Trichoderma reesei* SCNU-F0042 (Figure 1 and Appendix A). Eventually, under the guidance of molecular networking, three new dimeric sorbicillinoids (**1**–**3**) and one new 3,4,6-trisubstituted *α*-pyrone (**5**), as well as seven analogues (**4** and **6**–**11**) were isolated from the mangrove-derived fungus *Trichoderma reesei* SCNU-F0042 (Figure 2). Details of the isolation, structure elucidation, and bioactivities of these compounds are reported herein.

## 2. Results

14-hydroxybislongiquinolide (**1**) was yellowish amorphous powder. Its molecular formula was determined as C_28_H_32_O_9_ by ^13^C NMR and negative-ion HRESIMS data (*m*/*z* 511.1975, [M-H]^−^, calcd for C_28_H_31_O_9_, 511.1974), indicating 13 indices of hydrogen deficiency. The ^1^H NMR data (Table 1, Appendix A) of **1** showed resonances of five methyls [*δ*_H_ 1.89 (3H, d, *J* = 6.1 Hz), 1.45(3H, s), 1.38(3H, s), 1.18(3H, s), and 0.99(3H, s)], one oxygenated methylene [*δ*_H_ 4.22(2H, d, *J =* 4.4 Hz), three methines [*δ*_H_ 3.36(H, d, *J =* 5.6 Hz), 3.34(H, s), and 3.22(H, d, *J =* 5.6 Hz)], and eight olefinic protons [*δ*_H_ 7.35(H, dd, *J* = 14.5, 11.2 Hz), 7.20(H, dd, *J* = 15.3, 10.7 Hz), 6.55–6.59(H, m), 6.43–6.49(H, m), 6.38–6.42(H, m), 6.31–6.35(H, m), 6.25–6.29(H, m), and 6.18(H, d, *J =* 15.3 Hz)]. The ^13^C NMR data (Table 1, Appendix A) revealed 28 carbon resonances corresponding to four carbonyls (*δ*_C_ 210.3, 203.1, 198.3, and 179.1), five methyls (*δ*_C_ 24.1, 23.3, 19.1, 11.4, and 6.4), one oxygenated methylene (*δ*_C_ 63.0), three methines (*δ*_C_ 52.2, 43.9, and 43.6), twelve olefinic C-atoms (*δ*_C_ 185.5, 168.6, 147.9, 144.9, 143.0, 142.5, 131.7, 129.9, 128.8, 121.6, 111.7, and 94.7), and three quaternary (*δ*_C_ 84.8, 75.9, and 63.7). These NMR data were similar to those of bislongiquinolide (**4**), which was previously isolated from the fungi *Trichoderma longibrachiatum* Rifai aggr., indicating that they had the same core skeleton structure [18]. The major difference was the replacement of the C-14 methyl group of the alkene in the side chain in bislongiquinolide by an hydroxymethyl (*δ*_C/H_ 63.0/4.22) in **1**, which was further supported by the key ^1^H−^1^H COSY correlations (Figure 3 and Appendix A) of H-10/H-11/H-12/H-13/H_2_-14 and HMBC correlations (Appendix A) from H_2_-14(*δ*_H_ 4.22) to C-12(*δ*_C_ 129.9), C-13(*δ*_C_ 143.0) and from H-16(*δ*_H_ 6.18), H-17(*δ*_H_ 7.20) to C-15(*δ*_C_ 203.1). To determine relative and absolute configurations of compound **1**, the method of nuclear overhauser effect spectroscopy (NOESY) correlations (Figure 4), coupling constants (Table 1), circular dichroism (CD) spectra (Figure 5), and biogenetic considerations were used. The *E* geometries of double bonds about Δ^10^, Δ^12^, Δ^16^, and Δ^18^ were deduced based on the coupling constants of H-11 (*J* = 14.5, 11.2 Hz) and H-17 (*J* = 15.3, 10.7 Hz). In addition, the NOE correlations (Figure 4 and Appendix A) of 5-CH_3_(*δ*_H_ 1.18)/H-10(*δ*_H_ 6.43–6.49) and H-4(*δ*_H_ 3.34)/H-10(*δ*_H_ 6.43–6.49) suggest Δ^3^ was the *Z*-type and 5-CH_3_ and H-4 were located on the side chain from C-9 through C-14. The NOESY correlations of H-10(*δ*_H_ 6.43–6.49)/21-CH_3_ (*δ*_H_ 1.38) and H-4(*δ*_H_ 3.34)/21-CH_3_ (*δ*_H_ 1.38) were oriented H-4 and 21-CH_3_ to the side chain from C-9 to C-14, which is the same as 5-CH_3_. The NOESY correlations of 1-CH_3_ (*δ*_H_ 0.99)/H-17 (*δ*_H_ 7.20) and H-17(*δ*_H_ 7.20)/23-CH_3_ (*δ*_H_ 1.45) indicated that the same orientation of 1-CH_3_ and 23-CH_3_ with another side chain from C-7 through C-20. The 7*R**, 8*S** relative configuration was suggested by the coupling constant (*J*_7,8_ = 5.6 Hz) [18]. Thus, the relative configuration of compound **1** should be 1*R**, 4*S**, 5*S**, 7*R**, 8*S**, 21*S**, which adopted the same configuration of the bicyclo [2.2.2] octanedione core as bislongiquinolide (**4**) [18]. The absolute configurations of compound **1** proposed to the absolute configurations were 1*R*, 4S, 5*S*, 7*R*, 8*S*, 21*S*, which were the same as bislongiquinolide (**4**) comparisons of the information between them, including the similar optical rotation values, the biosynthetic pathway, the trend in CD curves (Figure 5), and the chemical shifts [19]. Hence, compound **1** was identified as the 14-hydroxylated analogue of bislongiquinolide (**4**), and named as 14-hydroxybislongiquinolide.

20-hydroxybislongiquinolide (**2**) was isolated as a yellowish amorphous powder. The molecular formula of **2**, the same as **1**, was determined as C_28_H_32_O_9_ by HRESIMS in observing a protonated molecular ion at *m*/*z* 511.1975 [M-H]^−^ (calcd for C_28_H_31_O_9_, 511.1974). In addition, the hydrogen protons and carbons of **1** and **2** were also of the same type by comparison of the NMR data. However, the position of the hydroxymethyl group was changed from C-14 to C-20, which was confirmed by the key ^1^H−^1^H COSY correlations (Figure 3 and Appendix A) of H-16/H-17/H-19/H-19/H_2_-20, as well as HMBC correlations (Appendix A) from H_2_-20(*δ*_H_ 4.22) to C-18(*δ*_C_ 128.5), C-19(*δ*_C_ 147.6) and from H-17(*δ*_H_ 7.25), H-16(*δ*_H_ 6.32) to C-15(*δ*_C_ 202.5). The relative configuration of **2** was the same as **1** based on their similar NOESY correlations (Figure 4). Together, they shared the same ECD Cotton effect at 226, 280 and 320 nm of **2** in the experimental ECD spectrum, which was identical to that of **1** (Figure 5). Thus, the absolute configuration of **2** was identified as 1*R*, 4S, 5*S*, 7*R*, 8*S*, 21*S*. The structure of **2** was established and named as 20-hydroxybislongiquinolide.

14, 20-dihydroxybislongiquinolide (**3**) was obtained as a yellowish amorphous powder and it was analyzed by HRESIMS (*m*/*z* 527.1925 [M−H]^−^, calcd. 527.1923) for the molecular formula C_28_H_32_O_10_, which had an extra oxygen atom than **1**. Its ^1^H and ^13^C NMR data (Table 1, Appendix A) were very similar to those of **1**, with the exception that the methyl signals (C-20, *δ*_C/H_ 19.1/1.89) in **1** were replaced by hydroxymethyl signals (C-20, *δ*_C/H_ 63.0/4.22). This deduction was also confirmed by the key ^1^H−^1^H COSY correlations (Figure 3 and Appendix A) of H-10/H-11/H-12/H-13/H_2_-14 and H-16/H-17/H-18/H-19/H_2_-20. The relative configuration of **3** was also the same as **1** deduced from the NOESY correlations (Figure 4). At the same time, the experimental ECD of **3** also has a negative Cotton effect at 226 and 320 nm, as well as positive Cotton effect at 280 nm, just like that of **1** (Figure 5). Therefore, the absolute configuration of **3** was identified as 1*R*, 4S, 5*S*, 7*R*, 8*S*, 21*S* and **3** was determined as 14, 20-dihydroxybislongiquinolide.

Acetylchrysopyrone B (**5**), a yellowish amorphous powder, has a molecular formula of C_13_H_12_O_6_ determined by HREIMS data (*m*/*z* 263.0560 [M-H]^−^; calcd for C_13_H_11_O_6_, 263.0561), with 8 degrees of unsaturation. The ^1^H NMR spectral data (Table 2, Appendix A), along with the HSQC data, informed the presence of two methyls [*δ*_H_ 2.34(3H, s) and 1.91(3H, s)] and five olefinic protons [*δ*_H_ 7.37(1H, dd, *J* = 15.2, 11.5 Hz), 7.15(1H, dd, *J* = 15.2, 11.5 Hz), 6.70 (1H, d, *J* = 15.2 Hz), 6.28 (1H, d, *J* = 15.2 Hz), and 6.47(1H, s)]. The ^13^C NMR and HSQC spectra (Appendix A) displayed 13 carbon resonances, including two carbonyls (*δ*_C_ 167.9 and 163.6), one carboxyl (*δ*_C_ 167.4), eight olefinic C-atoms (*δ*_C_ 158.9, 155.6, 143.4, 132.2, 130.2, 126.6, 115.5, and 106.6) and two methyls (*δ*_C_ 20.8 and 10.3). The above spectroscopic features suggested that **5** had a close structural relationship to chrysopyrone B [20], and the only difference was the appearance of the acetyl group connected to 4−OH, which was confirmed by the HMBC correlations from H_3_-8 to C-4 and C-7, and from H_3_-3 to C-4 and C-7 (Figure 3 and Appendix A). The geometry of double bonds Δ^1′^ and Δ^3′^ were deduced based on the coupling constants of H-2′ and H-3′ (*J* = 15.2, 11.5 Hz), suggesting the *E*-type of D1′ and D3′ double bonds. Therefore, the structure of **5** was proposed and named acetylchrysopyrone B.

The structures of the known compounds, bislongiquinolide (**4**) [18], saturnispol H (**6**) [21], sorbicillin (**7**) [22], trichodimerol (**8**) [23], bisorbicillinolide (**9**) [24], saturnispol B (**10**) [21], and bisvertinolone (**11**) [25], were recognized by comparing with their spectroscopic data reported in the literature. In terms of biological activity, the known compounds **4** and **6**–**11** have been evaluated for antibacterial [18,21,26], anti-inflammatory [14], radical scavenging activity [24], and PPAR*γ* agonist [27], etc.

Biological activities of all new compounds were evaluated in virus bioassays. The results showed that compound **2** displayed moderate antiviral activity against SARS-CoV-2, the causative agent of COVID-19 infection, as assessed in 293T cells, exhibiting an EC_50_ value of 29.0 μM, while compounds 1, 3, and 5 displayed no activity at 80 μM. All the tested compounds showed no cytotoxicity at 80 μM. Remdesivir (IC_50_ = 1.2 μM) was used as a positive control (Table 3).

## 3. Experimental Section

### 3.1. General Experimental Procedures

Optical rotations were recorded on an Anton Paar (MCP 500) polarimeter at 25 °C (Graz, Austria), and ECD spectra were acquired on an Applied Photophysics Chirascan spectropolarimeter (Surrey, UK). HRESIMS spectra were obtained on a ThermoFisher LTQ−Orbitrap−LCMS spectrometer (Palo Alto, CA, USA). A Bruker AVANCE NEO 600 MHz spectrometer (Bruker BioSpin, Rheinstetten, Germany) was used to record the 1D and 2D NMR using TMS as an internal reference at room temperature. Column chromatography (CC) was performed on silica gel (100–200 and 200–300 mesh; Qingdao Marine Chemical Factory, Qingdao, China) and Sephadex LH-20 (25–100 μm; GE Healthcare Bio-Sciences AB, Stockholm, Sweden). HPLC analysis uses a Waters 2695 system (Waters, Milford, MA, USA) with an ACE Excel 5 C18-AR column (250 × 4.6 mm, 5 μm; Hichrom Limited, Leicestershire, UK).

### 3.2. Fungal Material

The fungal strain *Trichoderma reesei* SCNU-F0042 was isolated from the fresh bark of the mangrove plant *Bruguiera gymnorhiza* collected from Qi’ao Island Mangrove Nature Reserve, Zhuhai City, Guangdong Province, China. The fungus was obtained using the standard protocol for isolation. The sequence data of the fungal strain have been deposited at Gen Bank with accession no. OP978317. A BLAST search result showed that the sequence was the most similar (99%) to the sequence of *Trichoderma reesei* (compared to OK445677.1). A voucher strain was deposited in School of Chemistry, South China Normal University, Guangzhou, China, with the access code *Trichoderma reesei* SCNU-F0042.

### 3.3. General Experimental Procedures

Spores of the fungal strain were inoculated into solid autoclaved rice medium in 400 1L Erlenmeyer flasks, each of which contained 50 g rice and 50 mL 0.3% sea salt, culturing in room temperature under static condition for 30 days. The mycelia and solid rice medium were soaked with MeOH and extracted with EtOAc three times. The organic solvents were evaporated under 48 °C with reduced pressure and obtained 158.6 g of organic crude extract. The extract was isolated by column chromatography over silica gel eluting with a gradient of PE/EA (1:0−0:1) to yield 5 fractions (Frs. 1−5). Fr. 3 (2.87 g) was subjected to Sephadex LH−20 (DCM/MeOH *v*/*v*, 1:1) to afford four sub-fractions (SFrs. 3.1−3.4). SFr. 3.2 (674.2 mg) was applied to silica gel CC (DCM/MeOH *v*/*v*, 60:1.5) to give compound **4**(28.1 mg) and compound **5** (8.2 mg). Fr. 4 (3.62 g) was subjected to Sephadex LH−20 (DCM/MeOH *v*/*v*, 1:1) to afford four sub-fractions (SFrs. 4.1−4.3). SFr. 4.2 (641.3 mg) was applied to silica gel CC (DCM/MeOH *v*/*v*, 60:2) to give compound **7** (5.1mg)**,** compound **9** (4.4 mg) and compound **11** (7.6 mg). Fr. 5 (1.82 g) was subjected to Sephadex LH−20 (DCM/MeOH *v*/*v*, 1:1) to afford three sub-fractions (SFrs. 5.1−5.3). SFr. 5.2 (582.1 mg) was applied to silica gel CC (DCM/MeOH *v*/*v*, 60:4) to afford two sub-fractions (SFrs. 5.2.1 and SFrs. 5.2.4). SFrs. 5.2.1 (85.3 mg) was further purified by Sephadex LH-20 CC eluted with MeOH to give compound **1** (8.2 mg). SFrs. 5.2.2 (68.1 mg) was further purified by silica gel CC (DCM/MeOH *v*/*v*, 60:3) to give compound **2** (4.3 mg). SFrs. 5.2.3 (56.5 mg) was further purified by silica gel CC (DCM/MeOH *v*/*v*, 60:3.5) to give compound **6** (4.6 mg) and compound **8** (4.2 mg). Fr. 6 (1.43 g) was subjected to Sephadex LH−20 (DCM/MeOH *v*/*v*, 1:1) to afford four sub-fractions (SFrs. 6.1−6.4). SFr. 6.2 (321.7 mg) was applied to silica gel CC (DCM/MeOH *v*/*v*, 60:4.5) to afford three sub-fractions (SFrs. 6.2.1—SFrs. 6.2.3). SFrs. 6.2.2 (32.5 mg) was further purified by Sephadex LH-20 CC eluted with MeOH to give compound **3** (3.1 mg). SFrs. 6.2.3 (46.0 mg) was further purified by silica gel CC (DCM/MeOH *v*/*v*, 60:4) eluted with MeOH to give compound **10** (6.1 mg)

### 3.4. Spectral and Physical Data of Compounds ***1**–**3*** and ***5***

14-hydroxybislongiquinolide (**1**): Yellowish amorphous powder; [α]_25_^D^ + 143 (c = 0.1, MeOH); UV (MeOH) *λ*_max_ (log *ε*) 294(4.36), 366(4.28) nm; IR (neat) *ν*_max_ 3380, 2965, 1735, 1661, 1452, 1442, 1381, 1258, 1200, 1152, 852, 802, 758 cm^−1^; ^1^H and ^13^C NMR data; see Table 1. HRESIMS: *m*/*z* 511.1975 [M-H]^−^ (calcd for C_28_H_31_O_9_, 511.1974).

20-hydroxybislongiquinolide (**2**): Yellowish amorphous powder; [α]_25_^D^ + 175 (c = 0.1, MeOH); UV (MeOH) *λ*_max_ (log *ε*) 291(4.44), 369(4.36) nm; IR (neat) *ν*_max_ 3325, 2985, 1741, 1670, 1465, 1453, 1379, 1243, 1202, 1148, 856, 805, 747 cm^−1^; ^1^H and ^13^C NMR data; see Table 1. HRESIMS: *m*/*z* 511.1975 [M-H]^−^ (calcd for C_28_H_31_O_9_, 511.1974).

14, 20-dihydroxybislongiquinolide (**3**): Yellowish amorphous powder; [α]_25_^D^ + 150 (c = 0.1, MeOH); UV (MeOH) *λ*_max_ (log *ε*) 287(4.21), 369(4.06) nm; IR (neat) *ν*_max_ 3451, 2975, 1733, 1656, 1470, 1445, 1383, 1255, 1198, 1146, 870, 802, 751 cm^−1^; ^1^H and ^13^C NMR data; see Table 1. HRESIMS: *m*/*z* 527.1925 [M-H]^−^ (calcd for C_28_H_31_O_10_, 527.1923).

Acetylchrysopyrone B (**5**): Yellowish amorphous powder; [α]_25_^D^ + 174 (c = 0.10, MeOH); UV (MeOH) *λ*_max_ (log *ε*) 260(4.12), 355(4.07) nm; IR (neat) *ν*_max_ 3185, 1735, 1651, 1562, 1423, 1367, 1260, 1145, 1019, 995 cm^−1^; ^1^H and ^13^C NMR data; see Table 2. HRESIMS: *m*/*z* 263.0560 [M-H]^−^ (calcd for C_13_H_11_O_6_, 263.0561).

### 3.5. LC-MS/MS and Molecular Networking Analysis

The EtOAc extract of *Trichoderma reesei* SCNU-F0042 was analyzed by LC−MS/MS. In positive-ionization conditions (*m*/*z* 200−800), the mobile phase consisted of 1‰ HCOOH formic acid in H_2_O and CH_3_CN. The elution gradient conditions for the LC mobile phase were as follows, based on times (t): t = 0−1 min, hold at 90% H_2_O/CH_3_CN; t = 1−10 min, increased linearly to 40% H_2_O/CH_3_CN; t = 10−13 min, increased linearly to 10% H_2_O/CH_3_CN; t = 13−16 min, hold at 10% H_2_O/CH_3_CN; t = 16−16.2 min, increased linearly to 90% H_2_O/CH_3_CN; t = 16.2−20 min, hold at 90% H_2_O/CH_3_CN with the flow rate of 0.3 mL/min. A total of 1 μL of the sample (*c* 1 mg/mL, CH_3_CN) was injected. The MS/MS data of EtOAc extract was first saved as .mzML format files through MSConvert software. The molecular networking was performed using the GNPS data analysis workflow and the spectral clustering algorithm. Parameters for molecular network generation were set as follows: precursor mass tolerance *m*/*z* 2.0 Da, fragment ion tolerance *m*/*z* 0.5 Da, cosine score above 0.7, minimum matched fragment ions 6, minimum cluster size 2, network TopK10. Data were visualized using Cytoscape 3.8.2 software.

### 3.6. SARS-CoV-2 Inhibition Assay

#### 3.6.1. Cell Lines and Virus

HEK 293T-hACE2 (ATCC CRL-3216 derived) was described previously [28]. African green monkey kidney epithelial cell line Vero were obtained from The Cell Bank of the Chinese Academy of Sciences, CBCAS, Shanghai, China. HEK 293T-hACE2 and Vero were maintained in Dulbecco’s modified Eagle’s medium (Invitrogen, Carlsbad, CA, USA) containing 10% fetal bovine serum (FBS, GIBCO, Carlsbad, CAe), 2 mM *L*-glutamine, 100 μg/mL streptomycin and 100 units/mL penicillin (Invitrogen) at 37 °C under 5% CO_2_. BA.2 (GDPCC 2.00299) was obtained from Guangdong Center for Human Pathogen Culture Collection (GDPCC), Guangdong Provincial Center for Disease Control and Prevention. BA.2 were amplified in Vero cells.

#### 3.6.2. Plasmids

ACE2 packaging construct (GeneCopoeia, EX-U1285-Lv105) uses a cytomegalovirus (CMV) promoter to express ACE2 and bears a puromycin selection marker in the integrating cassette.

#### 3.6.3. RT-qPCR Analysis

HEK 293T-hACE2 cells were seeded in 12-well flat-bottom plate at a density of 1.2 × 10^4^ cells/well. After 24 h, cells were incubated in media consisting of BA.2 (MOI of 0.01) and different concentrations of each compound for 1 h at 37 °C. After the incubation, cells were washed with sterile phosphate-buffered saline (PBS) once and incubated with media mixed with different concentrations of each compound, respectively, for further 48 h. Total RNA of each well was extracted from the cell culture supernatant using LogPure Viral DNA/RNA Kit (Magen, Guangzhou, China). Reverse transcription and qPCR were performed with Detection Kit for Novel Coronavirus (2019-nCoV) RNA (DA0932, DAAN GENE, Guangzhou, China). Samples were read on the QuantStudio7 Flex real-time PCR detection system (Thermo Fisher Scientific, Shanghai, China). The qPCR was performed in duplicates for each sample, and results were calculated using 2^–ΔCT^, where CT is threshold cycle [29].

### 3.7. Cell Viability Assay

The test compounds at a serial final concentration of 50 to 1 µM were evaluated against HEK 293T-hACE2 using the MTT method. Tested cell lines were cultured in Dulbecco’s modified Eagle’s medium (DMEM) (Invitrogen, Carlsbad, CA, USA) supplemented with 5% fetal bovine serum (Hyclone, Logan, UT, USA), 2 mM L-glutamine, 100 mg/mL streptomycin, and 100 units/mL penicillin (Invitrogen). The cultures were maintained at 37 °C in a humidified atmosphere of 5% CO_2_.

## 4. Conclusions

In summary, three new dimeric sorbicillinoids (**1**–**3**) and one new 3,4,6-trisubstituted α-pyrone (**5**), together with seven analogues (**4** and **6**–**11**), were isolated from the cultures of the mangrove endophytic fungus *Trichoderma reesei* SCNU-F0042 under the guidance of MS/MS based on molecular networking. Compound **2** exhibited moderate inhibitory effects on anti-SARS-CoV-2 activity with an EC_50_ value of 29.0 μM without cytotoxicity observed. Our study enriched the structural and biological activity diversity of sorbicillinoids.

## Figures and Tables

**Figure 1 marinedrugs-21-00442-f001:**
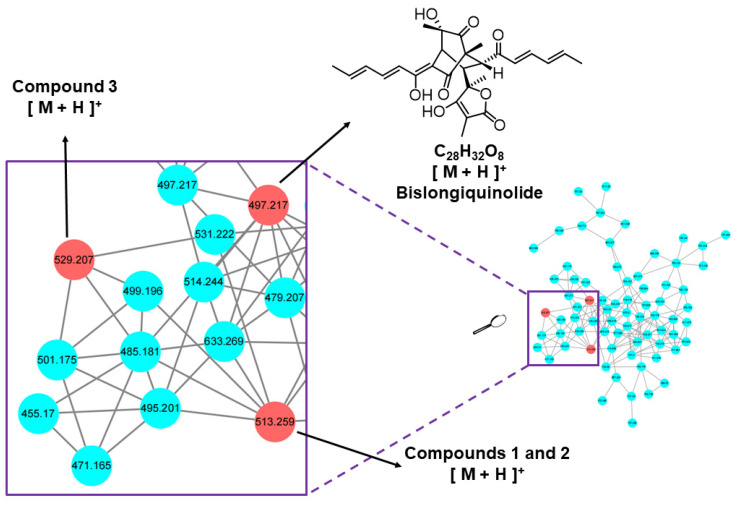
Cluster of nodes from *T. reesei* with compounds **1**–**4**.

**Figure 2 marinedrugs-21-00442-f002:**
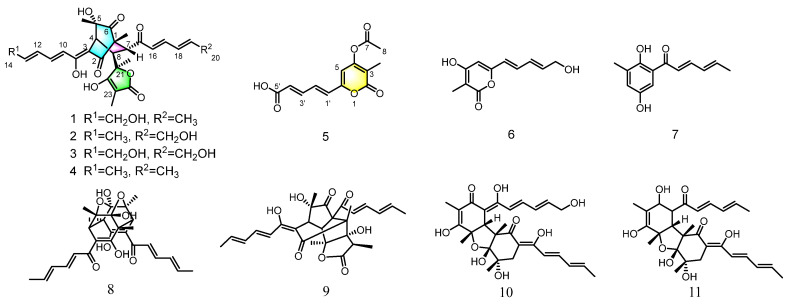
Structure of compounds **1**–**11**.

**Figure 3 marinedrugs-21-00442-f003:**
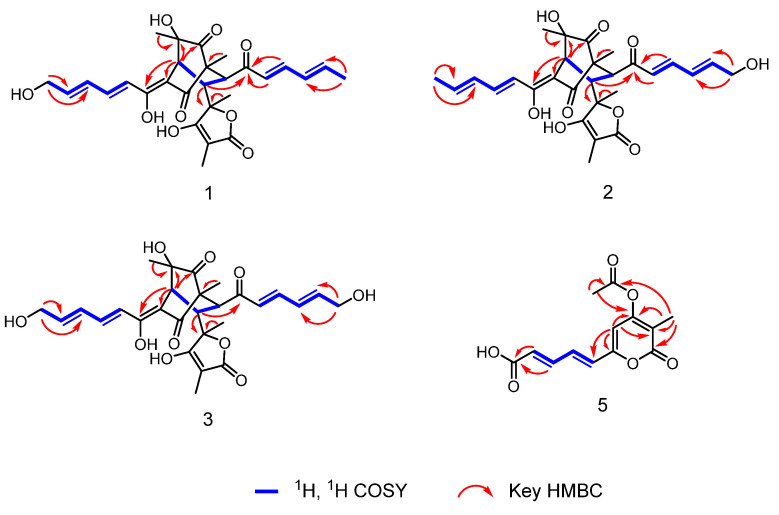
^1^H, ^1^H COSY, and key HMBC correlations of compounds **1**–**3** and **5**.

**Figure 4 marinedrugs-21-00442-f004:**
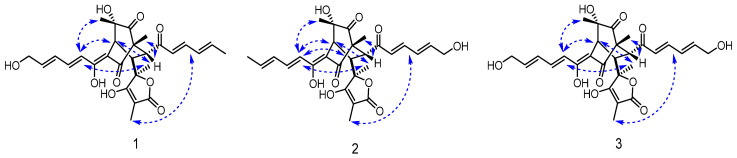
Key NOESY correlations of compounds **1**–**3**.

**Figure 5 marinedrugs-21-00442-f005:**
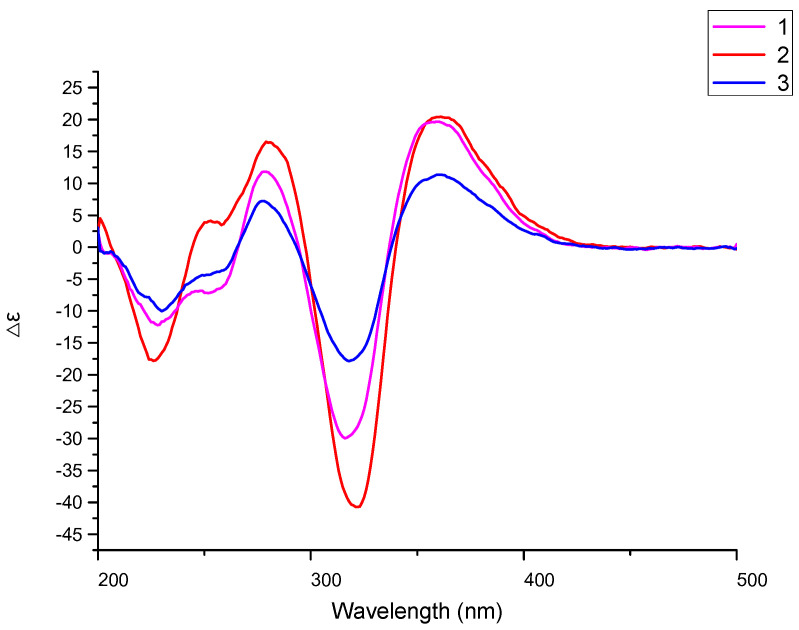
Circular dichroism (CD) spectra of compounds **1**–**3**.

**Table 1 marinedrugs-21-00442-t001:** ^1^H and ^13^C NMR data for compounds 1–3 (methanol-*d*_4_).

NO	1		2		3	
	*δ*_H_ (*J* in Hz)	*δ*_C_, Type	*δ*_H_ (*J* in Hz)	*δ*_C_, Type	*δ*_H_ (*J* in Hz)	*δ*_C_, Type
1	-	63.7, C	-	63.5, C	-	63.6, C
2	-	198.3, C	-	197.5, C	-	197.8, C
3	-	111.7, C	-	110.2, C	-	110.5, C
4	3.34, s	43.6, CH	3.33, s	43.5, CH	3.35, s	43.6, CH
5	-	75.9, C	-	75.8, C	-	75.8, C
6	-	210.3, C	-	210.2, C	-	210.2, C
7	3.36, d (5.6)	52.2, CH	3.31 ^a^	52.8, CH	3.33 ^a^	52.6, CH
8	3.22, d (5.6)	43.9, CH	3.28 ^a^	43.9, CH	3.26 ^a^	43.9, CH
9	-	168.6, C	-	169.5, C	-	169.4, C
10	6.43–6.49 ^a^	121.6, CH	6.28–6.33	119.6, CH	6.43–6.46	121.5, CH
11	7.35, dd (14.5, 11.2)	142.5, CH	7.30, dd (14.7, 11.2)	143.7, CH	7.37, dd (13.4, 11.5)	142.7, CH
12	6.55–6.59 ^a^	129.9, CH	6.39–6.43 ^a^	132.4, CH	6.52–6.60 ^a^	129.8, CH
13	6.25–6.29 ^a^	143.0, CH	6.19–6.25 ^a^	140.6, CH	6.26–6.32 ^a^	143.3, CH
14	4.22, d (4.4)	63.0, CH_2_	1.90, d (6.4)	18.9, CH_3_	4.23, d (4.1)	62.8, CH_2_
15	-	203.1, C	-	202.5, C	-	202.1, C
16	6.18, d (15.3)	128.8, CH	6.30–6.36 ^a^	130.0, CH	6.31–6.40 ^a^	130.1, CH
17	7.20, dd (15.3, 10.7)	147.9, CH	7.25, dd (15.4, 10.4)	146.8, CH	7.27, dd (15.2, 10.3)	146.6, CH
18	6.31–6.35 ^a^	131.7, CH	6.48–6.53 ^a^	128.5, CH	6.49–6.52 ^a^	128.6, CH
19	6.38–6.42 ^a^	144.9, CH	6.45–6.48 ^a^	147.6, CH	6.48–6.51 ^a^	147.5, CH
20	1.89, d (6.1)	19.1, CH_3_	4.22, d (4.0)	62.7, CH_2_	4.23, d (4.1)	63.0, CH_2_
21	-	84.8, C	-	84.6, C	-	84.6, C
22	-	185.5, C	-	182.0, C	-	182.9, C
23	-	94.7, C	-	96.2, C	-	96.1, C
24	-	179.1, C	-	178.2, C	-	178.2, C
1-CH_3_	0.99, s	11.4, CH_3_	0.99, s	11.3, CH_3_	0.99, s	11.3, CH_3_
5-CH_3_	1.18, s	24.1, CH_3_	1.18, s	24.2, CH_3_	1.19, s	24.2, CH_3_
21-CH_3_	1.38, s	23.3, CH_3_	1.40, s	23.4, CH_3_	1.33, s	23.3, CH_3_
23-CH_3_	1.45, s	6.4, CH_3_	1.48, s	6.5, CH_3_	1.47, s	6.4, CH_3_

^a^ Overlapped by other signals.

**Table 2 marinedrugs-21-00442-t002:** ^1^H and ^13^C NMR data for compound 5 (acetone-*d_6_*).

NO	*δ*_H_ (*J* in Hz)	*δ*_C_, Type
2	-	163.6, C
3	-	115.5, C
4	-	158.9, C
5	6.47, s	106.6, CH
6	-	155.6, C
7	-	167.9, C
8	2.34, s	20.8, CH_3_
1′	6.70, d (15.2)	130.2, CH
2′	7.15, dd (15.2, 11.5)	132.2, CH
3′	7.37, dd (15.2, 11.5)	143.4, CH
4′	6.28, d (15.2)	126.6, CH
5′	-	167.4, C
3-CH_3_	1.91, s	10.3, CH_3_

**Table 3 marinedrugs-21-00442-t003:** Results of anti-SARS-CoV-2 activity and cytotoxicity for 293T cell.

Compound	Inhibition of SARS-CoV-2 Viruses (EC_50_/μM) ^b^	Cytotoxicity for 293T Cell (IC_50_/μM) ^b^
**1**	NS ^c^	>80
**2**	29.0	>80
**3**	NS	>80
**5**	NS	>80
Remdesivir ^a^	1.2	>80

^a^ Positive control; ^b^ Data are shown as mean from three parallel experiments; ^c^ NS means not sensitive at 80 μM.

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
