# Peer review of "New Sorbicillinoids from the Mangrove Endophytic Fungus Trichoderma reesei SCNU-F0042"

_marinedrugs, 2023, doi:10.3390/md21080442_

Round 1

Reviewer 1 Report

Group of sorbicillinoids are relatively new compounds with a special structure having with different biological activities.

Authors present new derivatives of sorbicillinoids isolated from a mangrove endophytic fungus, a new trisubstituted alpha-pyron and its other analogues. The structure of the compounds were identified with wide-ranging and correct procedures. According the SARS-CoV-2 virus and 293T cell investigations authors established the moderate antivirus activity without any cytotocic action. Some of my comments are as follows:

1. lines 141-144: I suggest presenting the biological effects of similar compounds known so far cited here, 4, 6-11, if such studies are known. If so, it would be good to examine the new compounds for these bioassays as well.

2. Table 3: what is the explanation for the fact that only two biological tests were performed, especially in the case of cytotoxicity? It is difficult to draw conclusions from such a small number of studies

3. line 200: please correct a small mistake – correct: „…Compounds 1-3 and 5”

Author Response

We greatly appreciate reviewer’s comments and suggestions. We have revised the manuscript according to these comments. Revised portions are marked in yellow in the manuscript. All corrections in the paper and the responses to the reviewer comments please see the attached file.

Reviewer 2 Report

The manuscript “New Sorbicillinoids from the Mangrove Endophytic Fungus Trichoderma reesei SCNU-F0042” ly Long et al. reported the isolation and structural elucidation of three new  dimeric sorbicillinoids (1-3) and one new 3,4,6-trisubstituted α-pyrone (5) from the mangrove endophytic fungus Trichoderma reesei SCNU-F0042 using molecular networking approach. Compounds 2 showed moderate SARS-CoV-2 inhibitory activity with an EC50 value of 29.0 μM .

I recommend the manuscript will be accepted after minor revisions.

1.      Line 65. Add the dot after Rifai aggr.

2.      Line 78-79: add the NOESY correlations of H-4 and 21-CH3 as you have illustrated in figure 4.

3.      Line 137: Add the J values of H-2.

4.      Please provide the reference for SARS-COV-2 Inhibition Assay.

5.      Conclusion part: Line 271-272 and 276 Why did you mentioned the antimalarial activity here ?

Author Response

(The authors gave the same response as above.)
